# A 10-Year Retrospective Study on Pediatric Visceral Leishmaniasis in a European Endemic Area: Diagnostic and Short-Course Therapeutic Strategies

**DOI:** 10.3390/healthcare12010023

**Published:** 2023-12-21

**Authors:** Arianna Dondi, Elisa Manieri, Giacomo Gambuti, Stefania Varani, Caterina Campoli, Daniele Zama, Luca Pierantoni, Michelangelo Baldazzi, Arcangelo Prete, Luciano Attard, Marcello Lanari, Fraia Melchionda

**Affiliations:** 1Pediatric Emergency Unit, IRCCS Azienda Ospedaliero-Universitaria di Bologna, 40138 Bologna, Italy; arianna.dondi@aosp.bo.it (A.D.); luca.pierantoni@aosp.bo.it (L.P.); marcello.lanari@unibo.it (M.L.); 2Department of Medical and Surgical Sciences (DIMEC), University of Bologna, 40138 Bologna, Italy; stefania.varani@unibo.it; 3Specialty School of Paediatrics, Alma Mater Studiorum, University of Bologna, 40138 Bologna, Italy; elisa.manieri3@studio.unibo.it (E.M.);; 4Unit of Microbiology, IRCCS Azienda Ospedaliero-Universitaria di Bologna, 40138 Bologna, Italy; 5Infectious Diseases Unit, IRCCS Azienda Ospedaliero-Universitaria di Bologna, 40138 Bologna, Italy; caterina.campoli@aosp.bo.it (C.C.); luciano.attard@aosp.bo.it (L.A.); 6Pediatric and Adult CardioThoracic and Vascular, Oncohematologic and Emergency Radiology Unit, IRCCS Azienda Ospedaliero-Universitaria di Bologna, 40138 Bologna, Italy; michelangelo.baldazzi@aosp.bo.it; 7Pediatric Oncology and Hematology Unit, IRCCS Azienda Ospedaliero-Universitaria di Bologna S. Orsola Hospital, 40138 Bologna, Italy; arcangelo.prete@aosp.bo.it (A.P.); fraia.melchionda@aosp.bo.it (F.M.)

**Keywords:** visceral leishmaniasis, Liposomal Amphotericin B, pediatrics, leishmania, zoonoses

## Abstract

Background: Visceral leishmaniasis (VL) is a potentially fatal disease, with an increasing occurrence in northern Italy, affecting children and both immunocompetent and immunocompromised adults. Methods: This retrospective study conducted at the St. Orsola University Hospital of Bologna, Italy, evaluates the characteristics of 16 children (with a median age of 14.3 months) who were hospitalized between 2013 and 2022 for VL. Results: Seventy-five percent of patients presented with a triad of fever, cytopenia, and splenomegaly. An abdominal ultrasound examination revealed splenomegaly and hypoechoic spleen abnormalities in 93.8% and 73.3% of cases, respectively. Five VL cases were complicated by secondary hemophagocytic lymphohistiocytosis. Eleven patients were treated with a single 10 mg/kg dose of Liposomal Amphotericin B (L-AmB), while five received two doses (total of 20 mg/kg); one of the former groups experienced a recurrence. The fever generally decreased 48 h after the first L-AmB dose, and hemoglobin levels normalized within a month. The splenomegaly resolved in approximately 4.5 months. Conclusions: Pediatricians should consider VL in children with fever of an unknown origin, anemia, cytopenia, and splenomegaly. In our experience, abdominal ultrasounds and molecular tests on peripheral blood contributed to diagnosis without the need for bone marrow aspiration. The short-course therapy with two 10 mg/kg doses of L-AmB is safe and effective.

## 1. Introduction

Visceral leishmaniasis (VL), generally known as kala-azar, is a potentially lethal sandfly-borne disease caused by protozoans belonging to the *Leishmania donovani* complex. The two species that account for the majority of VL cases are *L. donovani*, which is mostly found in the Indian subcontinent and Eastern Africa, and *L. infantum*, which is found in the Mediterranean basin, Middle East, and Latin America [1]. Among tropical diseases, VL represents a large contributor to mortality and loss of disability-adjusted life years [2,3]; despite its substantial impact, the World Health Organization (WHO) considers VL to be one of the world’s neglected diseases [4]. VL is endemic in southern Europe, and while it typically occurs in rural areas, recent outbreaks in Europe have been reported in residential areas near urban centers in Madrid, Spain, and Bologna, Italy [5,6].

*L. infantum* VL is a zoonosis, and the main reservoir is found in domestic dogs, even though recent evidence suggests that other domestic and wild mammals may also be reservoirs, including cats, rodents, and hares [7]. Since dogs are the main reservoir, there is agreement on the importance of taking preventive measures against the spreading of *L. infantum*. In Europe, there is only one vaccine available for dogs, and its administration necessitates serological screening; moreover, in endemic areas, there is a common practice of conducting annual screenings for canine leishmaniasis [8]. No vaccine is available against human leishmaniasis, and control measures against sandflies are not standardized [9].

*L. infantum* VL can occur at any age, but it is more common in children, and the pathophysiology of the infection is characterized by a complicated interaction between the Leishmania strain and host characteristics [10]. VL’s clinical manifestations range from asymptomatic infections to severe diseases and, in some cases, fatal consequences [11]. After a subacute onset with constitutional symptoms, the classical clinical features are persistent irregular fever, marked hepatosplenomegaly, and weight loss. Normochromic normocytic anemia, thrombocytopenia, hypertransaminasemia, hypoalbuminemia, and hypergammaglobulinemia are common laboratory findings.

The diagnosis of VL is challenging and requires stepwise and multiple approaches [12]. Traditionally, diagnosis was confirmed via the direct demonstration of parasites in tissue specimens or cultures. Nevertheless, further diagnostic options have emerged, including rapid serological assays, such as the rK39-immunochromatographic test (rK39-ICT); additionally, Polymerase Chain Reaction (PCR) has garnered prominence as a diagnostic tool [13]. An abdominal ultrasound is part of the initial diagnostic work-up in patients with suspected VL, identifying splenomegaly, sometimes with distinctive multifocal hypoechoic nodules in pediatric patients [14]. A differential diagnosis is critical to distinguish VL from other illnesses, such as typhoid fever, tuberculosis, brucellosis, malaria, or other hematologic disorders [15], and to ensure proper treatment.

Disseminated Intravascular Coagulation (DIC) [16], hepatic failure, post-kala-azar dermal leishmaniasis, and secondary hemophagocytic lymphohistiocytosis (HLH) are among the most severe complications of VL [17,18]. HLH is a rare but potentially fatal complication caused by cytokine overproduction and the abnormal proliferation of cytotoxic lymphocytes and histiocytes, which results in hemophagocytosis. Indeed, it is well known that VL represents one of the infectious etiologies leading to secondary HLH as well as, for example, the Epstein–Barr Virus, Cytomegalovirus, Human Immunodeficiency Virus, and other infectious causes [15,19,20]. Due to its overlapping diagnostic features and onset in young children, it is critical to distinguish familial HLH (FHL) from secondary HLH induced by VL.

The treatment of VL is challenging due to several factors, including drug toxicity, resistance, epidemiological variability, and, most notably, a lack of evidence-based guidelines for the pediatric population [12,21].

The present study aims to describe clinical findings, laboratory characteristics, and outcomes in patients affected by VL caused by *L. infantum* and treated with Liposomal Amphotericin B (L-AmB) in our Centre.

## 2. Materials and Methods

The current study is an observational, retrospective, monocentric study.

All children aged between 0 and 14 years old, who had a microbiological diagnosis of VL and who were admitted and treated at St. Orsola University Hospital, IRCCS AOU Bologna, northeastern Italy, between 1 January 2013 and 31 December 2022 were included in this study. Patients who did not meet the inclusion criteria were excluded.

Data were retrieved retrospectively from the medical charts of children with VL who were discharged from the Pediatric Emergency Unit and the Pediatric Oncology and Hematology Unit. Each patient’s demographic and clinical data, area of residency, details about any stay in other VL-endemic areas, laboratory and microbiological investigations, abdomen ultrasonography abnormalities, therapy data, and clinical outcomes were recorded.

The microbiological diagnosis of VL was performed via serology and real-time PCR on bone marrow aspirate and/or peripheral blood samples as described [19]. In detail, from 2013 onwards, an rK39 ICT test was used in combination with other serological methods, including the indirect immunofluorescence assay, immunoenzymatic assay, or Western Blot. Concerning molecular diagnosis, DNA was amplified employing simultaneously two real-time PCR assays targeting the 18S ribosomal (18S r) RNA gene and the leishmanial kinetoplast DNA (kDNA) [22] until July 2022, while for cases identified after this date, a commercially available kit targeting the 18s gene was employed (*Leishmania* spp., Clonit, Abbiategrasso, Italy).

The study procedure is represented as a flowchart in Appendix A.

### 2.1. Statistical Analysis

Descriptive statistics of demographic and anamnestic data as ethnicity, place of residence, visiting VL endemic areas within 8 months before diagnosis, presence or contact with domestic animals, as well as clinical variables such as sex, age at diagnosis, the presence of symptoms, laboratory data and ultrasound results. Categorical and continuous variables were reported as the frequency with a relative percentage and median value with their range, respectively.

### 2.2. Ethical Statement

This retrospective chart review study is in accordance with the 1964 Helsinki Declaration and its later amendments. Informed consent was obtained by the patient’s guardians. The study was approved by the local Ethical Committee (protocol name LEISHMANIA-2019, protocol number 100/2020/Oss/AOUBo, approved in February 2020).

## 3. Results

During the study period (2013–2022), 16 pediatric patients (aged 0–14 years) were admitted to the Pediatric Emergency Unit or to the Pediatric Oncology and Hematology Unit at St. Orsola University Hospital, IRCCS AOU Bologna (northeastern Italy) diagnosed with VL. As shown in Table 1, the VL cases exhibited a median age of 14.3 months at diagnosis (range 4.3 months–5.5 years); 43.6% (7/16) of cases were male. Thirteen patients (81.3%) were younger than 24 months, and two children (12.5%) were less than 6 months old (Table 1). Nine patients (56.3%) lived in rural areas, while the remaining lived in urban areas. All children in the study were of Caucasian ethnicity.

Five patients (31.3%) reported visiting VL-endemic areas in the 8 months preceding the onset of symptoms, including Sicily (*n* = 2), the coast of Tuscany (*n* = 1), Albania (*n* = 1), and Morocco (*n* = 1). Seven (43.8%) children reported contact with dogs, one case denied contact, and the remaining eight (50.0%) had no information available. The average number of admissions per year was 1.6; however, 5 (31.3%) cases occurred in 2013. There were no cases reported in 2014, 2015, 2016, or 2019. All but one case (93.8%) developed symptoms between October and April; in one case, the onset of VL symptoms occurred in June (Figure 1).

In terms of clinical presentation, as shown in Table 1, all but one child (93.8%) had a fever, with a median body temperature of 39.7 °C. Splenomegaly occurred in 15 (93.8%) children, with a median spleen diameter of 9.3 cm (range 7.5–15.4). In addition, 11 out of 15 children (73.3%) had hypoechogenic areolae in the spleen as a distinctive ultrasound abnormality (Figure 2). Such lesions, characterized by a less defined echotexture, were better evaluated using high-frequency linear probes, which offer increased detection sensitivity in these cases. None of our patients had cutaneous lesions caused by *Leishmania*.

As shown in Table 1, the most frequent complete blood count abnormalities included anemia, which occurred in all children but one (15/16, 93.7%), and leukopenia and thrombocytopenia, both found in 81.2% of the patients (13/16). Pancytopenia was found in 12 out of 16 patients (75.0%). All the median values are reported in Table 2. All patients underwent an evaluation of their blood iron, and other causes of anemia were excluded.

Clinical, laboratory, and ultrasound findings are reported in Table 1; the median values and ranges of all laboratory findings are shown in Table 2.

The typical triad of the HLH syndrome (fever, cytopenia > 2 cell lines, and splenomegaly) was found in 12 (75.0%) cases. Among the overall study group, 5 (31.3%) children satisfied the definition of HLH, and a diagnosis was made by applying the revised diagnostic criteria of the HLH guidelines: the presence of at least 5 out of 8 diagnostic criteria allowed the diagnosis of HLH (Appendix A) [23].

No patients had immunodeficiency or other chronic disorders. Microbiological examinations of the pharyngo-nasal aspirates, blood, and feces revealed the presence of a concomitant viral infection at the time of admission in three (18.8%) cases, specifically Parvovirus B19, Influenza B virus, and Parainfluenza III virus. Two (12.5%) children developed a nosocomial infection via the Respiratory Syncytial Virus and Influenza A virus, respectively.

The median number of days required to establish a conclusive diagnosis since the onset of symptoms was 15.5 (4–80). More than 60 days elapsed between the onset of symptoms and the definitive diagnosis in one case due to mild and non-specific symptoms (isolated hypertransaminasemia). Another diagnostic delay was caused by an incorrect diagnosis of FLH. The median time from admission to diagnosis, on the other hand, was 4 (4–14) days. The patient with hypertransaminasemia was diagnosed before admission. The median length of total hospital stay was 15.5 (3–28) days, while the length of stay after treatment was 6.5 (2–14) days.

Concerning VL diagnosis, all five patients admitted in 2013 underwent bone marrow aspiration for molecular tests and a direct microscopic investigation. Starting in 2017, molecular analyses were performed on peripheral blood specimens; Leishmania DNA was detected by real-time PCR in all patients. One hundred percent of patients also tested positive at rK39 ICT.

Regarding treatment, eleven (68.8%) patients received a single dose of L-AmB totaling 10 mg/kg, whereas the remaining five (31.3%) patients received two doses on two consecutive days totaling 20 mg/kg, as shown in Table 3. Four (25.0%) children required an RBC transfusion. In our series, no adverse effects related to L-AmB infusions, such as chills, fever, nausea, vomiting, flushing, dizziness, and shortness of breath, occurred. Only three patients (18.8%) presented grade 1 hypokalemia (according to Common Terminology Criteria for Adverse Events, CTCAE v5), and levels of creatinine constantly within the normal range were observed. The median time for the fever to disappear was 48 (24–120) h.

Without considering the four transfused patients, 5/12 (41.7%) recovered 1 g/dL of hemoglobin at the time of discharge in roughly 7 (6–38) days. The average time for hemoglobin recovery was one (0.2–5) months. In our series, the only patient who relapsed had VL onset at 4 months of age and received as their initial treatment a single dose of L-AmB (10 mg/kg), reaching apyrexia within 48 h. The relapse occurred after 40 days with fever only; he was then treated with two consecutive doses of L-AmB 10 mg/kg, each with sustained resolution.

Among 15 patients with splenomegaly at diagnosis, an ultrasound follow-up was possible in 14 cases and performed up to 7 months after discharge. The ultrasound follow-up identified a complete remission in 10 out of 14 patients, achieved after a median time of 4.5 months (1–7 months), and a partial improvement of splenomegaly in 3 cases; only the relapsed patient (4 months old at diagnosis) showed splenomegaly after 8 months of follow-up. The disappearance of the hypoechoic splenic areolas required longer; they had improved but were still detectable in 6 out of 11 (54.5%) patients at the time of the last ultrasound check (4–8 months), whereas they had resolved in the remaining 5 cases.

Table 3 summarizes the patients’ features, therapy, and follow-up.

## 4. Discussion

The current study analyses the clinical features, laboratory findings, therapy, and outcomes in pediatric patients with VL caused by *L. infantum* and provides insights into our Clinical Centre’s experience with short-regimen L-AmB therapy.

We assessed the origin of the children from rural and urban areas, whether they travelled abroad in endemic areas during the previous months, and whether they had been in contact with dogs, considered the main reservoir. From our data, with the limit of the low number of cases, it does not seem that demographic and anamnestic data influenced the progression of the VL infection. VL is caused by *L. infantum* in Italy, including the Emilia-Romagna region in northeastern Italy. Between November 2012 and May 2013, an outbreak of VL was detected in the regional capital, Bologna; between January and May 2013, there were four of the five reported pediatric cases of VL in 2013, and the fifth of that year was in June [6]. The cause of this outbreak remains uncertain, although a connection to ongoing climate change has been hypothesized, as raised temperature and dryness in northern Italy are likely responsible for the recent increase in sandfly vectors in the same area [6,24,25]. Unlike other Italian regions, such as Sicily, where VL cases occur consistently year-round with no seasonal fluctuations [26], our area experienced a concentration of cases during the winter months, which is expected as the sandfly transmission period in the Mediterranean area peaks in July–September [27] and VL exhibits a long incubation period (2–6 months) [28].

The low median age of our cohort is consistent with other studies conducted on pediatric patients with VL in the Mediterranean area, which show a high prevalence of this infectious disease in young children [26,29,30]. This finding is most likely related to the immaturity of their cell-mediated immune system as well as to the absence of preexisting immunity. In particular, in our case series, the only patient who experienced a VL relapse was the youngest, as he was 4 months old; no demographic or anamnestic data, except for age, were related to VL relapse.

Our patients presented clinical and laboratory features comparable to those of other cohorts described in the literature [26,29], with the classic triad of fever, splenomegaly, and cytopenia present in more than 90% of patients, and anemia was the most frequent laboratory finding [29]. Most of our patients (11/16) exhibited elevated levels of ferritin and lactate dehydrogenase upon diagnosis as a result of the complex interplay between the host immune response, tissue and cellular damage, and inflammation caused by the parasite.

About two-thirds of our patients showed focal subcentrimetric hypoechoic splenic lesions, which have been shown to be frequent during ultrasound examinations in pediatric VL [14,31]; this ultrasound signature can be considered a useful diagnostic tool to suspect *L. infantum* infection in children. However, in the presence of these splenic abnormalities, alternative differential diagnoses, such as infectious, lymphoproliferative, or congenital conditions, should be investigated. Importantly, the presence of splenic abnormalities does not indicate a poor prognosis for patients with VL [32].

Regarding hematological alterations, hemoglobin levels normalized relatively quickly, with nearly half of the patients who did not receive blood transfusions showing an improvement at the time of discharge and complete normalization within a few months after the therapy, which is consistent with findings from other studies [29,30]. In contrast, as previously reported, the resolution of the ultrasonographic findings required longer times, and the disappearance of the hypoechoic splenic lesions proved to be a slower process than the resolution of splenomegaly [31]. Nevertheless, it is challenging to compare our data with other groups due to the small size of our cohort, variability in the timing of follow-up ultrasound scans, and limited available data on post-hospitalization follow-up [31,33]. According to a recent narrative review on ultrasonography findings in VL, splenomegaly can persist in a minority of patients for up to 8–10 months following treatment, while the clearance of splenic nodules has been reported to occur within 2–4 weeks or a few months after treatment [33].

The first step for VL diagnosis was carried out by serological screening; different tests were employed, always including rk39 ICT. All VL patients in our cohort tested positive for rk39 ICT, suggesting the higher sensitivity of this rapid test in the pediatric population compared to the modest sensitivity that was recently observed for the same test in adult VL [34]. All patients in our series had their diagnoses subsequently confirmed by molecular analyses. Numerous targets have been described to detect Leishmania DNA by real-time PCR; given their highly conserved nature and high copy number in the genome of the parasite, kDNA and 18S rRNA gene are the most commonly used for the diagnosis of VL [35]. Bone marrow aspirate is considered an elective specimen; however, real-time PCR using the above-mentioned targets showed comparable results on peripheral whole blood [36]. After the first wave of VL cases in 2013, diagnostic tools were improved at the Unit of Microbiology in our Clinical Centre, and bone marrow aspiration was no longer required; starting in 2017, VL diagnosis in pediatric cases was reached via positive Leishmania DNA detection in peripheral blood samples. The confirmation of VL by employing peripheral blood specimens plays a significant role in the diagnostic approach to pediatric VL, considering that bone marrow aspirate is an invasive procedure, which is performed under sedation in pediatric patients. Recent retrospective studies in Europe and South America showed that bone marrow aspiration was performed in most patients, ranging from 67% to 100% of the examined population [29,30,37,38]. In our experience, serologic and molecular VL diagnosis performed on peripheral blood is an efficient choice in pediatric patients. Nevertheless, despite the promising results emerging from our study, it is important to underline the adoption of a stepwise approach according to which the employment of microscopy and molecular tests on bone marrow aspirate can be evaluated in the case of PCR-negative or doubtful cases on peripheral blood samples and the high clinical suspicion of VL.

One of the patients in our study was initially treated with FLH-specific medications, meeting the diagnostic criteria (listed in Appendix A) [23]. Following early treatment failure and the negative results of genetic testing for FLH, the case was re-evaluated, and positive tests for *L. infantum* were documented. After discontinuing FLH therapy and receiving L-AmB treatment, the patient recovered. In line with the literature, a significant proportion of our patients met the clinical criteria for HLH but responded optimally to antiparasitic therapy [19,30]. Consequently, for proper therapy, it is crucial to consider the possibility of VL in pediatric patients fulfilling HLH criteria and to avoid immunosuppressive agents as well as more intensive treatments [15,29].

The WHO issued consensus recommendations in 2010 that recognized L-AmB as the first-line treatment for VL caused by *L. infantum* without providing precise guidance on the various dosing regimens [39]. In the same year, an open-label study was conducted on 412 patients in the Indian subcontinent in a 3:1 ratio to receive either a single dose of 10 mg/kg of L-AmB or conventional therapy (amphotericin B deoxycholate in 15 infusions). This trial revealed that using a single dose of L-AmB against *L. donovani* is a safe, effective, and non-inferior strategy [40]. Following this study by Sundar et al., from 2013 to 2019, VL patients treated at our Clinical Centre received a single dose of L-AmB totaling 10 mg/kg.

Given that VL is mostly caused by *L. infantum* in Europe, which necessitates a higher dosage of L-AmB for successful eradication, in 2017, the WHO recommended a therapy that included L-AmB at a proposed dosage of 3–5 mg/kg/day administered over 3–6 days, for a total cumulative dose of 18–21 mg/kg [28]. Recent studies, however, have highlighted the effectiveness of short treatment protocols using two consecutive daily doses of L-AmB at 10 mg/kg [41,42,43]. In a 2003 study, Syriopoulou and colleagues investigated the cost-efficacy of this short therapy in children through an open prospective study, comparing short to longer-lasting regimens with L-AmB (4 mg/kg daily for 5 days) and to pentavalent antimony therapy; the results showed a faster resolution of signs and symptoms in patients receiving the shorter L-AmB regimen [41]. Similarly, Krepis and colleagues examined the same short treatment regimens in children in 2017 and reported successful outcomes [42].

From 2020 onwards, we switched the pediatric VL treatment to a regimen based on two doses administered on two consecutive days for a total dose of 20 mg/kg, in line with evidence from the literature, highlighting the need for higher dosages of *L. infantum* VL compared to *L. donovani* [41]. Recent clinical trials in Europe have demonstrated that short-term therapy with two doses of 10 mg/kg of L-AmB is more effective than previous regimens, leading to rapid clinical responses, fever remission, and the normalization of blood findings [41,42].

In terms of clinical outcomes, our patients achieved the remission of fever and a reduction in inflammatory markers quickly after the first dose of L-AmB. The short therapy regimen we applied implies a higher initial dose (10 mg/kg) compared to longer therapeutic regimens, which may be responsible for the rapid response observed in pediatric patients. In contrast, patients treated with longer dosing schedules, such as 4 mg/kg daily for 5 days, experienced a relatively longer time for clinical improvement, as evidenced by the works of Syriopoulou et al. [41] and Krepis et al. [42] in Greece. This is most likely due to the drug’s pharmacokinetics, which favors higher initial dosages for better tissue penetration and longer visceral persistence over repeated lower doses [44].

One out of eleven patients in the group that received a single dose experienced a recurrence. Nevertheless, the VL relapse was effectively treated with the subsequent administration of two more doses of L-AmB at 10 mg/kg, each without any adverse effect. This patient’s retreatment posed no safety concerns and led to complete recovery. As mentioned, it is noteworthy that this patient was the youngest of the study and only 4 months old. In our opinion, VL relapse in this patient could be reasonably related to the young age and the sub-optimal dose of L-AmB (10 mg/kg) received as their initial therapy.

Eventually, all the other patients were treated effectively and achieved the complete resolution of the disease, which is in agreement with studies analyzing the efficacy of L-AmB for VL treatment in our geographical area, where resistance is uncommon [42].

There were no significant adverse reactions to the therapy in any of the pediatric patients, whether they received a single dose or two doses of L-AmB. This suggests the safety profile of high-dose L-AmB, as documented in the literature [41,42].

A therapeutic alternative is the use of miltefosine, an oral drug developed initially as an antineoplastic agent, which has been demonstrated to be a teratogenic agent. A prolonged half-life, as well as the increased resistance of *L. infantum* against this drug, have led to its limited clinical application [45,46]. Consequently, it is not employed in the clinical practice of our center.

Our study presents some limitations that should be disclosed. First, the retrospective nature of the study poses a constraint. As a result, there is the possibility that some side effects of high-dose short-course therapy might not have been fully considered despite our full attention to any possible adverse reactions. Second, the modest sample size may influence the significance of the findings. Third, this is a single-center study; thus, further multicenter analyses are needed to confirm our findings. However, the strength of this study relies on the long (10-year) observation period.

## 5. Conclusions

VL is classified as one of the “neglected tropical diseases” by the WHO. In Italy, this disease is spreading to previously non-endemic areas, showing a multiannual outbreak in the northeastern part of the country. VL caused by *L. infantum* significantly affects pediatric populations, emphasizing the need for pediatricians to consider VL in the differential diagnosis of cases of fever of unknown origin, the presence of HLH diagnostic criteria, and concurrent features such as anemia or cytopenia and splenomegaly. Notably, ultrasound findings of subcentimetric hypoechogenic areolae in the spleen should raise suspicion of VL. Given the recent availability of sensitive techniques for Leishmania DNA detection via real-time PCR, it is advisable to prioritize non-invasive methodologies by employing peripheral blood samples instead of bone marrow aspirates for VL diagnosis in the pediatric population. Especially in children, early diagnosis and prompt treatment are crucial to mitigate potentially severe complications such as secondary HLH and the need for hemotransfusion. In a setting where a pediatric standardized treatment protocol has not yet been established, the experience of our Clinical Centre might be of advice for pediatricians to safely avoid invasive diagnostic procedures, indicating the potential for a short-term high-dose L-AmB therapeutic approach to be effective in cases of VL caused by *L. infantum*. However, it is important to note that these findings are preliminary and require further validation through well-designed clinical trials.

## Figures and Tables

**Figure 1 healthcare-12-00023-f001:**
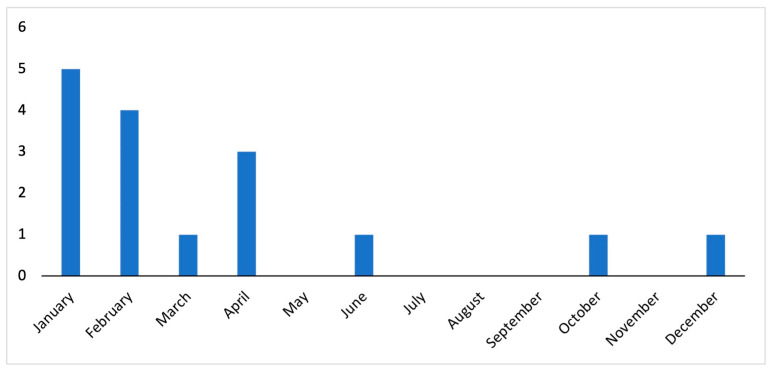
Monthly distribution of 16 pediatric VL cases referred to the St. Orsola University Hospital, Bologna, northeastern Italy, in the years 2013–2022.

**Figure 2 healthcare-12-00023-f002:**
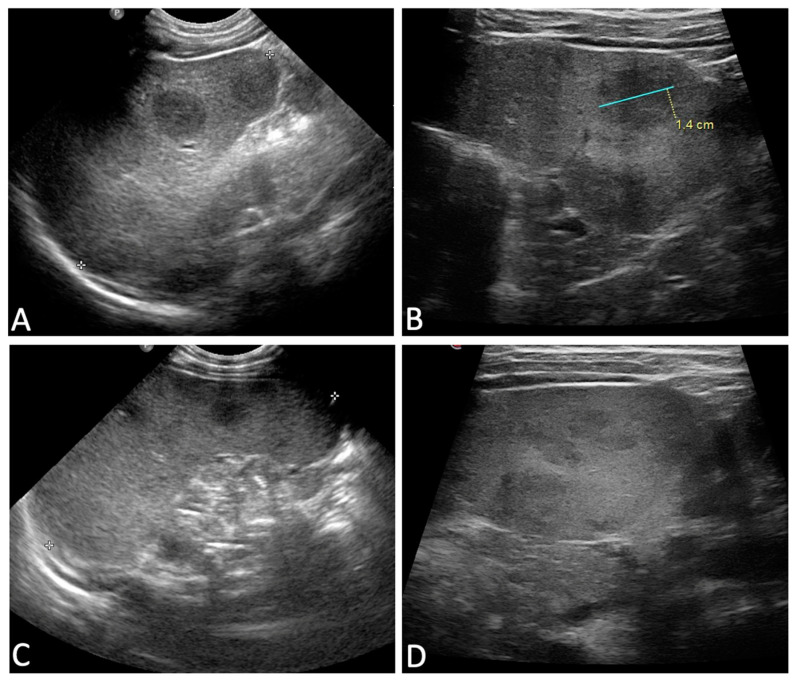
Variable ultrasonographic presentation of splenic lesions in VL. (**A**,**B**): large lesions with more clearly defined borders when compared to the surrounding healthy parenchyma. (**C**,**D**): small lesions that appear mildly hypoechoic in comparison to the surrounding parenchyma. (**A**,**C**) were taken with a convex probe, which is useful to better identify spleen dimensions and bigger lesions. (**B**,**D**) were taken with high-frequency linear probes, which are crucial, especially in the case of smaller lesions (**D**) that have a less defined echotexture, for which these probes offer increased detection sensitivity.

**Table 1 healthcare-12-00023-t001:** Demographic, clinical, laboratory, and ultrasound findings at admission.

Demographic, Clinical, Laboratory, and Ultrasound Findings	No of Patients (%)
Gender	
Male	7/16 (43.8)
Female	9/16 (56.2)
Age	
<18 months	11/16 (68.7)
>18 months and <24 months	2/16 (12.5)
>24 months	3/16 (18.8)
Area of residence	
Rural area	9/16 (56.2)
Urban area	7/16 (43.8)
Clinical Findings	
Fever	15/16 (93.8)
Diarrhea	3/16 (18.8)
Anorexia	1/16 (6.3)
Asthenia	7/16 (43.8)
Bilateral later cervical lymphadenopathy	1/16 (6.3)
Laboratory Findings	
Anemia	15/16 (93.8)
Thrombocytopenia	13/16 (81.3)
- Mild thrombocytopenia (100,000/mmc–150,000/mmc)	5/13 (38.5)
- Moderate thrombocytopenia (50,000/mmc–100,000/mmc)	8/13 (61.5)
Leukopenia	13/16 (81.3)
Pancytopenia	12/16 (75.0)
Elevated transaminases	9/16 (56.2)
Hypergammaglobulinemia	4/16 (25.0)
Ultrasound findings	
Splenomegaly	15/16 (93.8)
Splenic hypoechogenic areolae	11/15 (73.3)
Hepatomegaly	10/16 (62.5)

**Table 2 healthcare-12-00023-t002:** Median values (range) of the laboratory findings at admission.

Laboratory Findings	Median Value (Range)
Hemoglobin (g/dL)	8.0 (6.5–9.7)
Leukocyte count (/μL)	3910 (2590–7640)
Platelet count (/μL)	96,000 (40,000–457,000)
Serum Creatinine (mg/dL)	0.28 (0.16–0.49)
Proteins (g/dL)	6.45 (4.7–7.4)
AST/ALT (U/L)	106 (29–2177)/56.5 (10–1976)
LDH (U/L)	855.5 (234–1959)
CRP (mg/L)	3.12 (13.97–0.11)
Ferritin (ng/mL)	890 (12–13,362)
Triglycerides (mg/dL)	241.5 (124–374)
Fibrinogen (mg/dL)	222 (100–415)

AST: aspartate aminotransferase; ALT: alanine aminotransferase; LDH: lactate dehydrogenase; CRP: C-Reactive Protein.

**Table 3 healthcare-12-00023-t003:** Patients’ treatment and follow-up.

**Variables**	**Value**
Timing	
Median time from onset of symptoms to diagnosis (range)	15.5 days (4–80 days)
Median time of hospitalization (range)	15.5 days (3–28 days)
Median time of hospitalization after therapy (range)	6.5 days (2–14 days)
Diagnosis	
Bone marrow aspiration performed	5 (31.3%) patients
Bone marrow aspiration not performed	11 (68.7%) patients
Treatment	
L-AmB single dose (10 mg/kg)	11 (68.8%) patients
L-AmB double dose (20 mg/kg)	5 (31.2%) patients
Remission of fever after therapy	
<48 h	13 (81.2%) patients
>48 h	3 (18.8%) patients
RBC transfusion	
Yes	4 (25.0%) patients
No	12 (75.0%) patients
Relapse	
Yes	1 (6.3%) patient
No	15 (93.7%) patients

L-AmB: Liposomal Amphotericin B; RBC: red blood cells.

## Data Availability

The data presented in this study are available on request from the corresponding author.

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
