# Peer review of "A 10-Year Retrospective Study on Pediatric Visceral Leishmaniasis in a European Endemic Area: Diagnostic and Short-Course Therapeutic Strategies"

_healthcare, 2023, doi:10.3390/healthcare12010023_

Round 1

Reviewer 1 Report

Comments and Suggestions for Authors

Dear Editor

Thank you for the review opportunity. 

These concerns must be resolved before consideration:

In the method section it is stated that “The current study is an observational, retrospective, monocentric study, including 82 children, aged 0 – 14 years old” however the children were mostly 24 months. The data must be clear and correct in both parts.

In the result section, why the total population is not mentioned?, 16 out of…? In a long-term study how many children were assessed?

The method section is not in parallel with the result part.

According to the type of the study as retrospective, what inclusion and exclusion criteria was considered.

How the quality of the samples were controlled and adopted during the long duration of study? Were the tests done at a same time or were differently analyzed? If they were different were the same assays and kits applied?

Were there any justification according to the time of sampling?

Were the lifestyles of the children affected on the infection progression? Was this issue assessed? For instance the rural and urban areas.

The study procedure should be presented better if a flowchart is applied.

Best Regards.

Reviewer 2 Report

Comments and Suggestions for Authors

Reviewer 3 Report

Comments and Suggestions for Authors

This is a very good paper which aptly describes the presentation and evolving nature of visceral leishmaniasis in Europe.

I will add that the introduction could be more reflective of the lifecycle in Europe including strategies to reduce the stray dog population and vaccines which have been approved in canines. It would add to discuss the vector control methods presently being employed in Europe to combat this troublesome condition. 

Moreover, would be useful to add the ethnicity of the underlying patients as there is some variation to presentation and treatment response. Would be also good if you discuss the use of miltefosine in this arena as this is a therapeutic option. For your HLH diagnosis, was this per bone marrow or through Hscore ?

Comments on the Quality of English Language

Very good. 

Round 2

Reviewer 1 Report

Comments and Suggestions for Authors

Dear Editor

The changes are satisfying.

Best Regards.

Comments on the Quality of English Language

Dear Editor

The changes are satisfying.

Best Regards.